

# *Vernicia fordii* leaf extract inhibited anthracnose growth by downregulating reactive oxygen species (ROS) levels *in vitro* and *in vivo*

Luyao Ge[1,2], Yanling Zeng[1,2], Xinyun Liu[1,2], Xinhai Pan[1,2], Guliang Yang[3], Qinhui Du[1,2] and Wenlin He[1,2]

[1] Central South University of Forestry and Technology, Key Laboratory of Cultivation and Protection for Non-Wood Forest Trees, Ministry of Education, Changsha, Hunan, China

[2] Central South University of Forestry and Technology, Key Lab of Non-Wood Forest Products of State Forestry Administration, Changsha, Hunan, China

[3] Central South University of Forestry and Technology, National Engineering Laboratory for Rice and By-products Processing, Changsha, Hunan, China

Corresponding author
Yanling Zeng, zengyan-ling110@csuft.edu.cn, zengyanling110@126.com

## ABSTRACT

**Background**. *Colletotrichum fructicola* is a predominant anthracnose species in *Camellia oleifera*, causing various adverse effects. Traditional intercropping *Vernicia fordii* with *C. oleifera* may enhance anthracnose resistance, but the mechanism remains elusive.

**Methods**. We utilized UPLC-MS/MS and acid-base detection to identify the major antimicrobial alkaloid components in the *V. fordii* leaf extract. Subsequently, by adding different concentrations of *V. fordii* leaf extract for cultivating *C. fructicola*, with untreated *C. fructicola* as a control, we investigated the impact of the *V. fordii* leaf extract, cell wall integrity, cell membrane permeability, MDA, and ROS content changes. Additionally, analysis of key pathogenic genes of *C. fructicola* confirmed that the *V. fordii* leaf extract inhibits the growth of the fungus through gene regulation.

**Results**. This study discovered the alkaloid composition of *V. fordii* leaf extract by UPLC-MS/MS and acid-base detection, such as trigonelline, stachydrine, betaine, and O-Phosphocholine. *V. fordii* leaf extract successfully penetrated *C. fructicola* mycelia, damaged cellular integrity, and increased ROS and MDA levels by 1.75 and 2.05 times respectively, thereby inhibiting *C. fructicola* proliferation. By analyzing the key pathogenic genes of *C. fructicola*, it was demonstrated that the antifungal function of *V. fordii* leaf extract depends mainly on the regulation of *RAB7* and *HAC1* gene expression. Therefore, this study elucidates the mechanism of *V. fordii* -*C. oleifera* intercropping in strengthening anthracnose resistance in *C. oleifera*, contributing to efficient *C. oleifera* cultivation.

## INTRODUCTION

*Vernicia fordii,* a biomass energy tree prevalent in China, has been distributed in 16 provinces and internationally in places such as America, Argentina, and Paraguay since

the early 20th century (*Zhang et al., 2019*). *V. fordii* is a deciduous specious species that produces abundant fallen leaves in fall and winter (*Zhang et al., 2014*). Leaves of *V. fordii* possess bactericidal properties. Liu Liping's research (*Liu et al., 2009*) indicates that *V. fordii* leaf extract (VFLE) exhibits potential antibacterial activity against *Staphylococcus aureus*, *Escherichia coli*, and *Shigella dysenteroae*. 80% ethanol extract from *V. fordii* leaf showed superior bacteriostasis. However, the components responsive to antimicrobial activity, mode of action, and inhibitory properties of fungi remain unknown.

*Camellia oleifera*, a valuable oil plant with a high monounsaturated fatty acid content, is of concern due to anthracnose causing failure rates of up to 50% and plant decline (*Zeng et al., 2014*; *Li et al., 2016*). In oil tea-producing regions, anthracnose reduces seed yield by 10–30% annually with extreme cases of 40–50% loss of yield, resulting in significant economic losses (*Cao et al., 2023*). The disease affects multiple plant organs and has prominent effects on various vegetables and fruits, including coffee, cocoa, avocado, mangos, mulberry, pistachio, and persimmon, so forth (*Prihastuti et al., 2009*; *Rojas et al., 2010*; *Sanders & Korsten, 2003*; *Kwarteng et al., 2023*; *Xie et al., 2010*; *Yan et al., 2011*; *Yang et al., 2011*). Mainly systemically applied fungicides such as carbendazim or metribuzin in China are inefficient due to a single mechanism of action causing fungicide resistance and environmental pollution. Moreover, these fungicides lack specific targeting of *C. fructicola* while showing broad-spectrum activity (*He et al., 2022*; *Ma & Li, 2023*).

In China, intercropping between *V. fordii* and *C. oleifera* trees dates back to Xu Guangqi's "Agricultural Policy Book" in the Ming Dynasty. Intercropping of *V. fordii* with *C. oleifera* trees significantly enhances *C. oleifera* tree productivity compared to monoculture, and the anthracnose incidence rate of *C. oleifera* in the intercropped forests is lower than that of monoculture *C. oleifera* forests. The peak anthracnose occurrence time in *C. oleifera* coincides with the leaf-shedding phase of *V. fordii*. In mixed *C. oleifera* and *V. fordii* forests, the deciduous leaves of *V. fordii* serve as a natural canopy, containing bioactive constituents that impede the outbreak of *C. oleifera* anthracnose. Research indicates that intercropping of maize and *Cynanchum bungei* (He-Shou-Wu) effectively reduces the incidence rates of rust disease, leaf spot, downy mildew on *Cynanchum bungei*, and brown spot on *Cynanchum bungei* compared to monoculture. Furthermore, the flavonoid content increased by 8.42% and 6.58%, polyphenol content increased by 15.87% and 13.97%, and total steroidal glycosides content increased by 10.07% and 9.45% respectively in the intercropped treatments. Additionally, the yield of tuberous roots also increased by 14.5% and 13.8% respectively. These results indicate that the intercropping system of corn with *Polygonum multiflorum* is beneficial for reducing disease incidence (*Shen et al., 2021*). This may provide one explanation for the lower incidence rate of anthracnose in mixed *C. oleifera* and *V. fordii* forests compared to pure *C. oleifera* forests. Therefore, we speculate that certain *V. fordii* leaf components may inhibit *C. fructicola* and contribute to *C. oleifera* anthracnose relief. In our study, UPLC-MS/MS was used to identify the metabolites and major antifungal components of VFLE. Antifungal testing using *C. oleifera* anthracnose as the target sought to examine the effect of VFLE on *C. oleifera* anthracnose. The antifungal mechanism was investigated through mycelium growth observation, mycelium surface morphology

analysis, mycelium cell wall integrity examination, mycelium cell membrane permeability assessment, mycelium pathogenicity assessment, and pathogenic gene expression analysis.

## MATERIALS AND METHODS

For this investigation, the certified local cultivar 'Huatong No. 1' of *V. fordii* (Hemsl.) Airy Shaw, derived from the germplasm conservation bank in Yongshun County, Hunan Province, was employed. Ten-year-old, uniform monocarpic plants were chosen and their leaves, flowers, and husks were preserved at −80 °C following collection. Analysis of leaf metabolite composition was initiated within one month of collection.

Test strain: *Colletotrichum fructicola*, the cause of anthracnose disease in *C. oleifera*, was provided by the Microbial Strain Conservation Experiment Center of Central South University of Forestry Science and Technology.

Leaves of *C. oleifera*: Immature leaves of *C. oleifera* were collected from the nursery on the roof of the Tree Building of Central South University of Forestry and Technology (Changsha, Hunan, China).

### Preparation of extracts from *V. fordii* flowers, leaves, and husks

The samples of paulownia flowers, leaves, and fruit shells were ground with liquid nitrogen (30 Hz, 1.5 min) using an MM 400 grinder to powder form. After adding 200 mL of 80% acidic ethanol (pH adjusted to 3.0 with HCl) to each powder sample, further soaking was allowed for 24 h. After 1 h of sonication (SB-120DTN; 120W, Dongkang, Tianjin, China), the extract was filtered (DP-01; 20W, Scientz, Ningbo, Zhejiang, China). The extraction was then evaporated using a rotary evaporator, and the remaining paste was dissolved in three mL 0.1% DMSO before filtering through a 0.45 μm organic filter membrane (*Liu et al., 2009*).

### Component identification (UPLC-MS/MS)

All chromatographic separations were carried out utilizing an ultra-performance liquid chromatography (UPLC) system coupled with an Agilent SB-C18 column (1.8 μm, 2.1 mm ×100 mm) (USA). The mobile phase consisted of solvent A (ultrapure water with 0.1% formic acid, v/v) and solvent B (acetonitrile), subjected to an elution gradient initiated at 5% solvent B and gradually increasing to 95% over 9 min, followed by a 1-min hold before decreasing to 5% and maintaining for 3 min. The flow rate was set at 0.35 mL/min, column temperature at 40 °C, and the injection volume was 4 μL.

The MS settings consisted of: an ESI source operating at a temperature of 550 °C, a mass spectrometer voltage of 5,500 V, and a gas curtain set at 206.8 kPa. CAD with high parameter settings was used in the triple quadrupole (QQQ) instrument, with each ion pair scanned and detected based on optimized declustering voltage and collision energy.

### Quantification of major alkaloids in leaves of *V. fordii*

Concentrate a sufficient volume of VFLE into methanol to achieve a detection solution with a concentration of 10 μg/mL, utilizing an HPLC method concerninga standard curve consisting of trigonelline, stachydrine, O-phosphocholine, and betaine. The alkaloid content of *V. fordii* leaves was monitored using this procedure.

HPLC parameters included: Shimadzu Corporation's Shim-pack XR-ODS III column (75 mm × 2.0 mm, 1.6 μm), a column temperature maintained at 35 °C, and a flow rate of 0.2 mL/min. The mobile phase consisted of solvent A (a 0.1% aqueous solution of formic acid) and solvent B (acetonitrile). An initial 5% B ratio was used for 6 min, rising linearly to 15%, followed by a 3-min equilibrium. Next, the B ratio dropped to 5% and remained steady for another 5 min.

### Inhibition of extraction on pathogenicity of *C. fructicola*

The pathogenicity of the test strains was determined by conducting a detached leaf inoculation assay on *C. oleifolia* leaves, as described by reference *Qian et al. (2009)*. Fresh and uniform-sized young leaves of *C. oleifolia* were collected and washed with distilled water. The leaves were then placed in Petri dishes lined with moist filter paper, with a piece of degreased cotton moistened with distilled water covering the leaf stalk. In the treatment group, a sterilized needle was heated using an alcohol lamp and cooled. Small holes were punctured on both sides of the *C. oleifolia* leaves, and in these holes, fungal mycelial plugs (6 mm in diameter) obtained from culture using 600 μg/mL extracts of *V. fordii* flowers, leaves, and fruit husks were inoculated. The control group consisted of six mm fungal mycelial plugs cultured on a standard PDA medium. The sealed Petri dishes were incubated at a temperature of 28 °C (GNP-9050; Sanfa, Shanghai, China). After 4 days of dark incubation at 28 °C, the leaves were removed, and the diameters of the fungal mycelial growth were measured using a vernier caliper with a cross measurement. Each treatment had three replicates. The average value was used to calculate the growth inhibition rate of the detached leaf lesions:

$$\text{Inhibition rate of lesion growth} = \frac{\text{Lesion area of control group} - \text{Lesion area of treatment group}}{\text{Lesion area of control group}} \times 100\%.$$

### Study of the anti-potency of VFLE against *C. fructicola*

VFLE at various concentrations was treated in the culture medium where the *C. fructicola* fungus was then inoculated. The impact of VFLE on the parasitic mycelium of *C. fructicola* was evaluated by quantifying colony growth. The study used a PDB medium prepared using 3.5 g of potato dextrose agar + 100 mL of distilled water, sterilized at 121 °C for 20 min. Once cooled, VFLE at final concentrations of 0, 100, 200, 400, 600, 800, and 1000 μg/mL, respectively, was added. Fungal cubes (6 mm) of *C. fructicola* were placed and the medium was then incubated at 28 °C. Each test involved three repeats.

### SEM observation

The mycelia of *C. fructicola* were cultivated in the presence of 600 μg/mL VFLE and subsequently subjected to sequential washing with phosphate buffer salt solution (PBS) solution on two occasions. Subsequently, the pellet was resuspended in PBS and subjected to centrifugation (5000 rpm, 4 °C, 5 min) to separate the supernatant. The recovered mycelial biomass was then fixed in a 3% glutaraldehyde solution at 4 °C for 24 h, after which it underwent thorough rinsing with sterile water for 20 min. Subsequently, the samples were subjected to graded ethanol dehydration, with sequential immersion in 30%, 50%, 70%, and 95% ethanol solutions for 20 min each, culminating in a final dehydration step in absolute ethanol for 45 min. The treated mycelia were subsequently freeze-dried,

gold-coated, and examined under a scanning electron microscope (SEM-6380LV; Yishu, Shanghai, China) (*Cai et al., 2014*). All experiments had three replicates.

## Evaluation of cell wall integrity and cell membrane permeability

One hundred milliliters of PDB were supplemented with 100, 200, 400, and 600 μg/mL of VFLE, respectively. Subsequently, a six mm mycelium mass of *C. fructicola* was added to the medium and cultivated for 3 d at 28 °C in an oscillating incubator with 160 rpm. Supernatant and mycelium were isolated to assess extracellular alkaline phosphatase (AKP) activity and cell wall integrity, respectively. Three replicates per treatment were performed.

To examine cell membrane permeability, 0.10 g of mycelium was weighed and steeped in one mL PBS (37 °C, 5 min) and then rinsed thrice. Electrical conductivity was determined using a conductivity meter (DDSJ-319L; INES, Shanghai, China). Three replicates were used for each treatment.

## Assessment of oxidative stress

An amount of 0.10 g of mycelium was weighed and assayed for intracellular ROS levels using the non-specific ROS probe DCF-DA (*Gemes et al., 2016*). Absorbance was measured at 450 nm (ELISA reader, F-4500; Hitachi) to determine ROS concentration. MDA content was detected following the procedures outlined in the MDA assay kit purchased from Jiangsu Enzyme Free Company (China), with three replicates per treatment.

## Cellophane penetration assay

Mycelia measuring 2 mm × 2 mm were excised from VFLE-cultured colonies of *C. fructicola* and transferred to cellophane-coated PDA plates. Plates were maintained for 48 h at 28 °C after which cellophane was removed. Three replicates per treatment were included.

## Gene expression analysis

The PDB medium was supplemented with VFLE at final concentrations of 0, 100, 200, 400, and 600 μg/mL, followed by the addition of six mm mycelial discs. Cultures were incubated at 28 °C and 160 rpm for 48 h before samples were subjected to RNA extraction. RNA was designated as CK A, 0.5A, 1A, 2A, 3A, and preserved at −80 °C. Custom RT-qPCR primers (Table S1) were designed for the detection of *HAC1*, *MKK1*, *RAB7*, *VAM7*, and *VPS39* gene expressions. qRT-PCR was conducted on the CFX96™ real-time PCR system (Bio-Rad, Hercules, CA, USA) with SYBR® Premix Ex Taq™ II (AG) using three replicates per treatment.

## Statistical analysis

Each experiment was repeated thrice. Mass spectrometry data were processed utilizing the software SCIEX Analyst version 1.6.3. Experimental data, including AKP enzyme activity, extracellular conductivity, malondialdehyde content, reactive oxygen species (ROS) activity, anthracnose lesion size in detached leaves of *C. oleifera*, and the size of *C. fructicola* colonies on alkaloid standard culture plates, were processed and statistically analyzed *via* Excel 2016, IBM SPSS Statistics for one-way ANOVA, Image J, and Origin software. The expression levels of the selected genes were calculated using the $2^{-\Delta\Delta Ct}$ method.

## RESULTS

### Characterization of principal components in *V. Fordii* leaves

UPLC-MS/MS analysis identified 571 metabolites, with the percentage contribution of bioactive components such as phenolic acids, flavonoids, lignans, coumarins, tannins, alkaloids, and terpenoids exceeding 50%. Alkaloids and flavonoids have shown anti-bacterial and anti-inflammatory properties. The pH of the *V. fordii* leaves oil, tested in water and ethanol, was found to be above 8.0, indicating mainly alkaline properties (Table 1).

### Characteristics of key alkaloids in *V. fordii* leaves

Morphosate targeted metabolomics analysis established the substantial presence of several active alkaloid metabolites such as trigonelline, stachydrine, betaine, and O-phosphocholine (Table 2).

To unearth the predominant alkaloids in the antifungal efficacy of VFLE, a comprehensive HPLC analysis was performed. As shown in Fig. 1, the VFLE showed four peaks corresponding to standard compounds: trigonelline, stachydrine, betaine, and O-phosphocholine, respectively.

Peak area-based methods were used for quantitative analysis of trigonelline, stachydrine, betaine, and O-phosphocholine, revealing that VFLE exhibited the highest stachydrine content at 13.455 mg/L (Table 3).

### *V. fordii* extracts inhibited mycelial growth

Observations in Figs. 2A and 2D illustrate that lesions on *C. oleifera* leaves treated with pathogenic fungi were reduced upon exposure to extracts of *V. fordii* flower, leaf, and husk, respectively, indicating a reduction in virulence of *C. fructicola*, thereby constraining lesion expansion. Importantly, VFLE decreased fungi infectivity to a significant extent, showing only 18.75% lesion area compared to the control group (Table 4). These findings suggest VFLE as an effective mitigant of *C. oleifera* anthracnose, laying the groundwork for future development.

The results of antifungal activity tests on *C. fructicola* by primary alkaloids from *V. fordii* are shown in Figs. 2B, 2E, and Table 5. Notably, all four alkaloid criteria significantly inhibited *C. fructicola*, with stachydrine and trigonelline showing superior suppression compared to betaine and O-phosphocholine, resulting in significantly reduced colony diameters compared to the control group. The efficacy experiment revealed reduced virulence of the fungus post-cultivation with alkaloid standards, yielding smaller lesions compared to untreated control (Fig. 2C). Specifically, stachydrine and betaine showed significantly higher inhibitory effects on lesion production than trigonelline and O-Phosphocholine, resulting in lesion areas only one-fourth the size of the control group (Fig. 2F). Thus, it appears that the primary bioactive alkaloids responsible for limiting *C. fructicola* growth in VFLE are trigonelline and betaine.

### VFLE inhibition concentration

VFLE's effect on *C. fructicola* growth, at various concentrations, is demonstrated in Fig. 3 and Table 6. *C. fructicola* mycelia ceased growth after 2 d steeped in a PDB liquid medium

**Table 1  Principal active components in the metabolism of *V. fordii* leaves.**

| Composition types | Composition | Characteristic component |
|---|---|---|
| Phenolic acid | Benzamide, Benzoic acid, 2-Phenylethanol, Salicylic acid*, Cinnamic acid, Protocatechuic acid*, Caffeic aldehyde, Terephthalic acid*, Coniferyl alcohol, Caffeic acid, Methyl 2,4-dihydroxyphenylacetate, Methyl ferulate, 2-Nitrophenol, 2,6-Dihydroxybenzoic acid, Anthranilic acid, Pyrogallol, Dibutyl phthalate*, Neochlorogenic acid*, Tetragallic acid, Ferulic acid, Methyl ferulate, Sinapinaldehyde, 2-Feruloyl-sn-glycerol, 1-Feruloyl-sn-glycero etc. 106 species in total. | Caffeoylnicotinoyl, Illoyltartaric acid |
| Flavone | Naringenin*, Phloretin, Phloretin-2′-O-(6′-O-xylosyl)glucoside Homoeriodictyol, Quercetin-3-O-robinobioside*, Quercetin*, Dihydroquercetin(Taxifolin), Catechin gallate*, 5,7,3′,4′,5′-Pentahydroxydihydroflavone, Gallocatechin-gallocatechin-gallocatechin, Isohyperoside*, Catechin gallate*, Quercetagetin-4′-Methyl Ether*, Robinetin*, 5,7,4′-Trihydroxyisoflavone-7-O-galactoside-rhamnose* Syringetin, Kaempferol-3-O-rhamnoside (Afzelin)(Kaempferin)*, Epicatechin gallate*, Naringenin-7-O-Rutinoside(Narirutin)*, Quercetin-3-O-glucoside (Isoquercitrin)*, Naringenin-4′-O-glucoside*, Dihydroquercetin(Taxifolin), Phloretin-2′-O-(6′-O-xylosyl)glucoside, Limocitrin (5,7,4′-trihydroxy-8,3′-dimethoxyflavone)* etc. 99 species in total. | — |
| Lignan | ClemaphenOl A, 4-Nitrocatechol, Pinoresinol-4-O-glucoside, Pinoresinol , Medioresinol-4′-O-(6′′-acetyl)glucoside, Pinoresinol-4-O-(6′-acetyl)glucoside etc. 9 species in total. | — |
| Coumarin | Methyl Brevifolincarboxylate, Scopoletin-7-O-glucuronide, Isofraxidin*, Scopoletin-7-O-glucoside (Scopolin), Isoscopoletin (6-Hydroxy-7-Methoxycoumarin), Fraxidin (8-Hydroxy-6,7-dimethoxycoumarin)* etc. 10 species in total. | — |
| Tannin | Ellagic acid, Geraniin, Gallic acid, Digallic Acid, Brevifolin, Gemin D, 1,3,6-Tri-O-galloyl-D-glucose*, 7-O-Galloyl-D-sedoheptulose, 3,3′,4-Trimethoxyellagic acid, 1,3,4,6-Tetra-O-Galloyl-D-Glucose, 2-O-Salicyl-6-O-Galloyl-D-Glucose, 1,2,3,4,6-Penta-O-Galloyl-D-Glucose etc. 31 species in total. | — |
| Alkaloid | Spermine, Trigonelline, Piperidine, N-Benzylmethylene isomethylamine, Tryptamine, Indole, Choline, Betaine*, Acetylcholine, Indole-3-carboxaldehyde, Histamine, Strictinin, Nicotianamine, *O-Phosphocholine*, trigonellin, Choline Alfoscerate, 6-Deoxyfagomine* etc. 39 species in total. | — |
| Terpenes | Ursonic acid*, Ketoursolic acid*, Camaldulenic acid, Pentadecanoic acid etc. 8 species in total. | 16 $\alpha$-Hydroxytrametenolic acid |

**Table 2  Principal alkaloid metabolite content in VFLE extract.**

| Sequence | name of compound | Molecular weight/Da | Lon mode | The molecular weight of parent ion/Da | molecular weight of ion/Da |
|---|---|---|---|---|---|
| 1 | Trigonelline | 137.048 | + | 138.05 | 94.07 |
| 2 | Stachydrine | 143.095 | + | 144.1 | 84.1 |
| 3 | Betaine | 117.079 | + | 118.09 | 59 |
| 4 | O-phosphocholine | 184.073 | + | 184 | 125 |

Notes.
"+" Indicates positive ion mode.

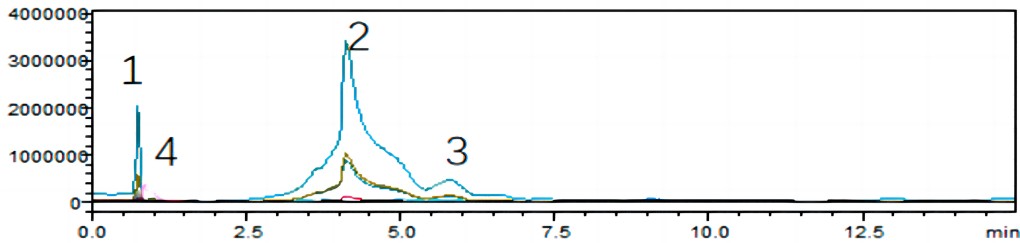

**Figure 1  HPLC chromatograms of VFLE extract.** Peaks 1: Trigonelline; Peak 2: Stachydrine; Peak 3: Betaine; Peak 4: O-phosphocholine.

**Table 3  Principal alkaloid content in VFLE extract.**

| Sequence | name of compound | Retention time | Qualitative ion pair | Regression equation | Compound content (mg/L) |
|---|---|---|---|---|---|
| 1 | Trigonelline | 0.92 | 118.1>42.1 | $Y = 170332X + 2.26225e+006$ | 3.874 |
| 2 | Stachydrine | 3.56 | 137.8>92.0 | $Y = 386728X + 1.06064e+006$ | 13.455 |
| 3 | Betaine | 5.79 | 145.1>58.0 | $Y = 26589.0X + 310872$ | 4.698 |
| 4 | O-Phosphocholine | 0.94 | 83.8>124.9 | $Y = 123731X + 1.16866e+006$ | 2.588 |

with VFLE concentration reaching 600 µg/mL (Fig. 3E). On further incubation for two more days on PDA plates with VFLE above 600 µg/mL, *C. fructicola* growth completely ceased at a concentration of 800 µg/mL (Fig. 3I). This illustrates VFLE's potent inhibition of *C. fructicola* growth and its positive antifungal effects.

## VFLE compromised the cell wall and membrane of *C. fructicola*

To examine the anti-fungal effects of VFLE on *C. fructicola*, cell membrane integrity and wall integrity were examined. Initially, *C. fructicola* mycelium subjected to various treatments was visualized by scanning electron microscopy. The surface morphologic alterations induced by VFLE are depicted in Fig. 4. The untreated control group presented a smooth, complete, and intact structure, showing even thickness and robust growth (Fig. 4A). Conversely, with 600 µg/mL VFLE, there were rough, irregular, and prominent wrinkles (Fig. 4B). Next, wall damage was estimated through AKP activity. The experiment

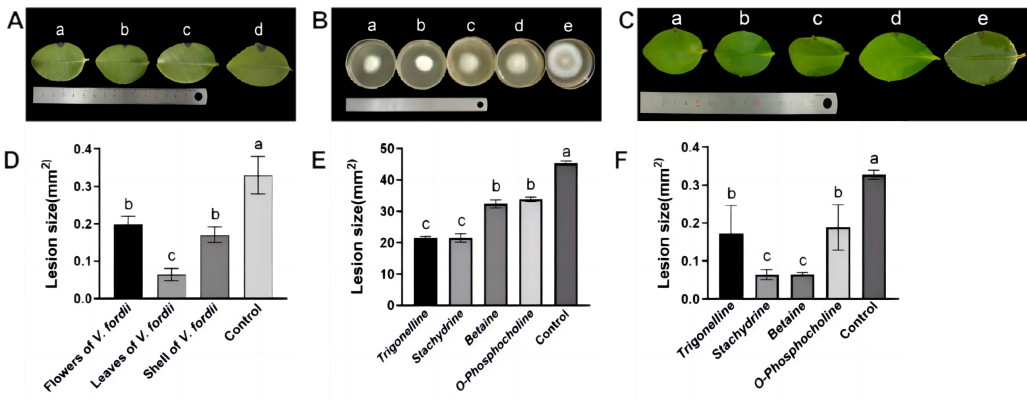

**Figure 2 Inhibition of growth of *C. fructicola* mycelium by *V. fordii* extracts.** (A) Map of *C. oleifera* lesions post-*V. fordii* treatment, treatments a-d: *V. fordii* application on flowers, leaves, and husks, and a control; (B) colony growth pattern with treatments a-e: Trigonelline, Stachydrine, Betaine, O-Phosphocholine, and a control; (C) *in vitro* leaf lesion growth; (D) spot sizing analysis for defining *C. oleifera* lesion pathogenicity; (E) colony diameter alteration; (F) lesion size modifications, treatments a-e: trigonelline, stachydrine, betaine, O-phosphocholine, and a control. The values represent the means ± standard deviations. Statistical analysis was performed using *t*-test. The lowercase letters indicate a significant difference at the 0.05 level, and the same letters indicate insignificant differences.

**Table 4 Antifungal activity of 600 µg/mL *V. fordii* extracts.** The values represent the means ± standard deviations. Statistical analysis was performed using t-test. The lowercase letters indicate a significant difference at the 0.05 level, and the same letters indicate insignificant differences.

| Extract | Diameter of colony /mm | Inhibition rate of colony growth /% | The disease of spot area /mm² | Inhibition rate of disease spot growth/% |
|---|---|---|---|---|
| Flower of *V. fordii* | 34.88 ± 0.26 b | −8.89 | 0.21 ± 0.19 b | 34.37 |
| Leaf of *V. fordii* | 23.73 ± 0.26 b | 25.91 | 0.06 ± 0.13 c | 81.25 |
| Husk of *V. fordii* | 22.61 ± 1.91 b | 29.40 | 0.18 ± 0.16 b | 43.75 |
| CK | 32.03 ± 0.36 ab | | 1.32 0.45 a | |

**Table 5 Inhibition of *V. fordii* extracts on the main alkaloid content of mycelium.**

| Alkaloid | Diameter of colony /mm | Inhibition rate of colony growth /% | The disease of spot area /mm² | Inhibition rate of disease spot growth /% |
|---|---|---|---|---|
| *Trigonelline* | 23.76 ± 0.48 c | 46.00 | 0.17 ± 0.25 b | 45.16 |
| *Stachydrine* | 23.92 ± 1.32 c | 46.17 | 0.06 ± 0.05 c | 80.64 |
| *Betaine* | 34.50 ± 1.36 b | 22.07 | 0.07 ± 0.02 c | 80.64 |
| *O-Phosphocholine* | 34.88 ± 0.78 b | 21.22 | 0.18 ± 0.21 b | 42.93 |
| Control | 46.35 ± 0.29 a | 0 | 1.32 ± 0.15 a | 0 |

**Notes.**
The values represent the means ± standard deviations. Statistical analysis was performed using *t*-test. The lowercase letters indicate a significant difference at the 0.05 level, and the same letters indicate insignificant differences.

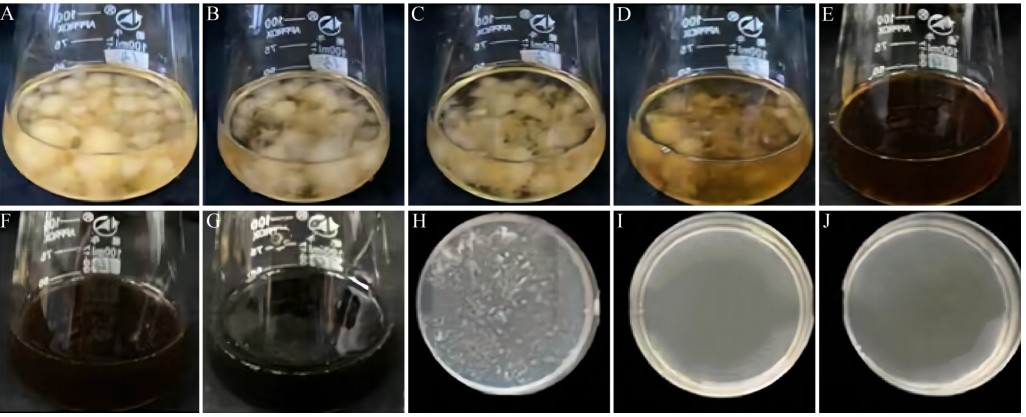

**Figure 3  Effect of VFLE on mycelial growth of *C. fructicola*.** (A–G) 0, 100, 200, 400, 600, 800, 1000 μg/mL VFLE on liquid culture of *C. fructicola*, respectively; (H–J) 600, 800, 1000 μg/mL VFLE on solid culture of *C. fructicola*, respectively.

**Table 6  Effect of *V. fordii* extracts on mycelial growth of *C. fructicola*.**

| Concentration (μg/mL) | 0 | 100 | 200 | 400 | 600 | 800 | 1000 |
|---|---|---|---|---|---|---|---|
| PDB Culture 2d | ++ | ++ | ++ | ++ | − | − | − |
| PDA Culture 2d | ++ | ++ | ++ | ++ | + | − | − |

**Notes.**
"++" indicates vigorous colony growth in the medium; "+" indicates small colony growth; "-" indicates no colony growth. Three biological replicates for each treatment.

showed dramatic augmentation in the VFLE-treated group, reaching 0.111 ± 0.005 U/mL, significantly higher than the untreated control group (0.07 ± 0.09 U/mL, $p < 0.05$) (Fig. 4C). This significant increase signifies compromised cell wall integrity, causing AKP release from the periplasmic space into the extracellular space. In addition, the cell membrane's response was evaluated by investigating relative electrical conductivity shifts. As shown in Fig. 4D, the treated group showed an appreciable increase in relative electrical conductivity compared to the untreated control group, and the conductivity escalated progressively with VFLE addition. Noteworthy, a sharp increase within the 200–400 μg/mL VFLE dose range was seen, subsequently reaching a plateau with escalating concentration, indicating saturated electrolyte leakage and slow conductivity increment.

## VFLE induced an increase in oxidative stress in fungal mycelia

Considerable increases in malondialdehyde (MDA) were observed in the VFLE-treated mycelium, as depicted in Fig. 5. Notably, the MDA levels escalated with elevated VFLE concentration, particularly at 200–400 μg/mL inducing substantial apoptosis of *C. fructicola* mycelium. Furthermore, the introduction of 100 μg/mL VFLE to the medium resulted in a marked rise in the mycelium's reactive oxygen species (ROS) content, suggesting the accumulation of ROS post-VFLE treatment in *C. fructicola* mycelium.

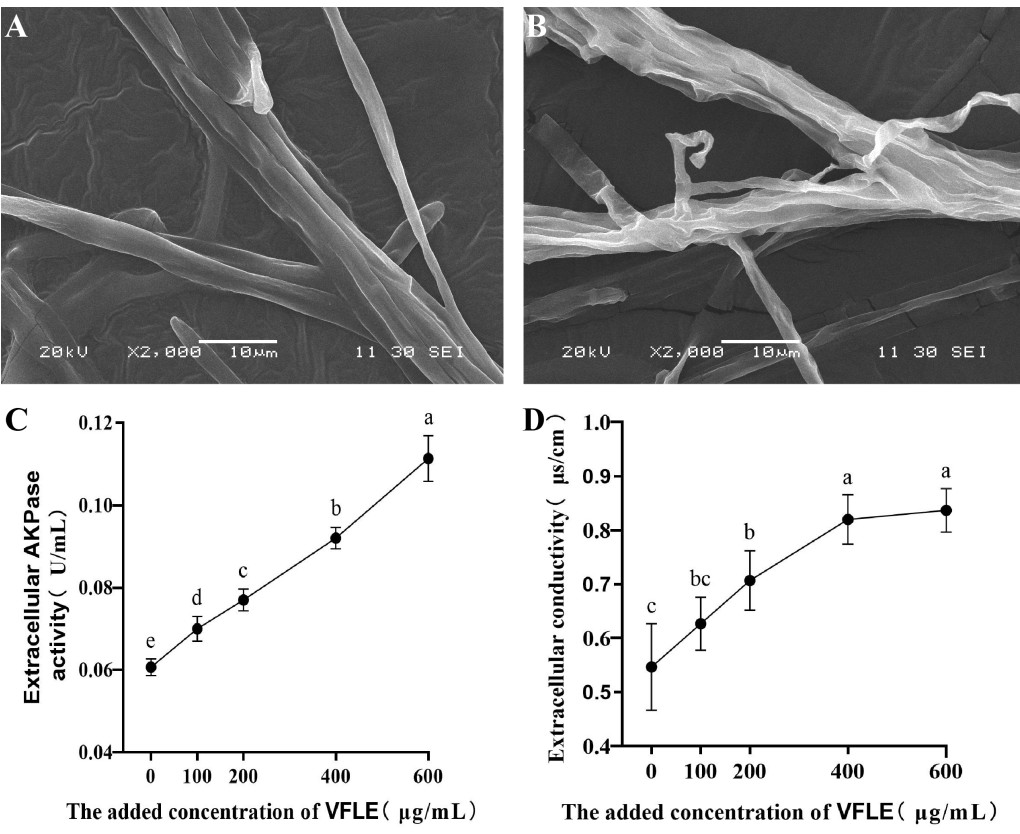

**Figure 4** **Impact of VFLE on cell wall integrity and membrane permeability of *C. fructicola*.** (A) *C. fructicola* mycelium in control group; (B) *C. fructicola* mycelium treated with 600 μg/mL; (C) AKP change under VFLE treatment; (D) alteration in electrolyte leakage content under VFLE treatment. The values represent the means ± standard deviations. Statistical analysis was performed using *t*-test. The lowercase letters indicate a significant difference at the 0.05 level, and the same letters indicate insignificant differences.

## VFLE influenced the colonization capacity of *C. fructicola*

Cellophane mimicry of *C. oleifera* served as a substrate on which cultures with varying levels of VFLE were inoculated. Observations demonstrated that post-cellophane removal, non-VFLE, and 100 and 200 μg/mL VFLE-treated fungi consistently formed colonies. Conversely, 400 and 600 μg/mL VFLE-treated strains failed to proliferate (Fig. 6), suggesting VFLE-restricted *C. fructicola* penetration.

## VFLE affected the expression of key pathogenic genes

The addition of varying concentrations of VFLE significantly altered the expression levels of five major pathogenic genes in *C. fructicola* (Fig. 7). These genes showed their peak expression at an extract concentration of 100 μg/mL, then gradually declined with increased extract concentration. This observation indicates that VFLE, at a concentration of 100 μg/mL, amplifies the pathogenicity of *C. fructicola* in *C. oleifera*. However, as VFLE concentration increases, *C. fructicola* growth in *C. oleifera* is inhibited, resulting in reduced

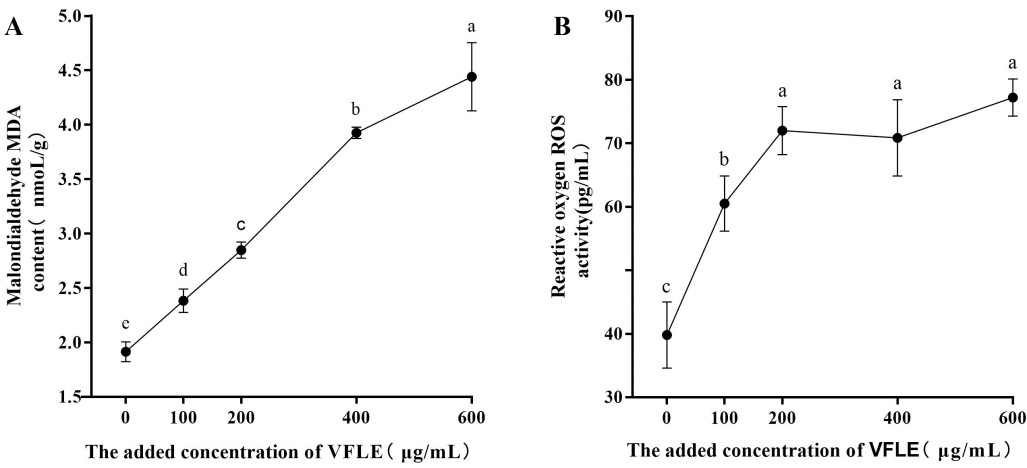

**Figure 5** **Effect of VFLE on ROS and MDA content of *C. fructicola*.** (A) Propylene glycol MDA content; (B) ROS activity. The values represent the means ± standard deviations. Statistical analysis was performed using *t*-test. The lowercase letters indicate a significant difference at the 0.05 level, and the same letters indicate insignificant differences.

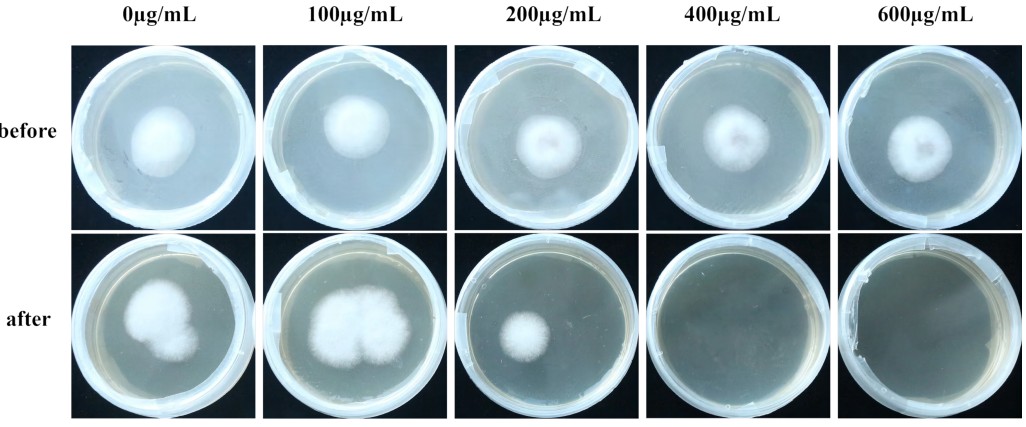

**Figure 6** **Effect of VFLE on penetration of *C. fructicola* cellophane.** Before: Cellophane membrane colonies; After: Cellophane membrane colonies removal.

pathogenicity. The most profound effect was observed for *RAB7*, followed by *HAC1* and *VPS39*.

## DISCUSSION

Rotation and intercropping strategies increase plant resistance to pathogens, enhance antifungal activity, and thus reduce disease incidence (*Han et al., 2016*; *Yu et al., 2019*). Intensive tea plantations benefited from the inclusion of climate-resilient crops (sorghum, cowpea) and aromatic volatile blends of semen cassiae, marigold, and flemingia as companion crops, mitigating disease and pest risks (*Pokharel et al., 2023*). Intercropping pest and disease suppression can be attributed to leaf and root extract components and

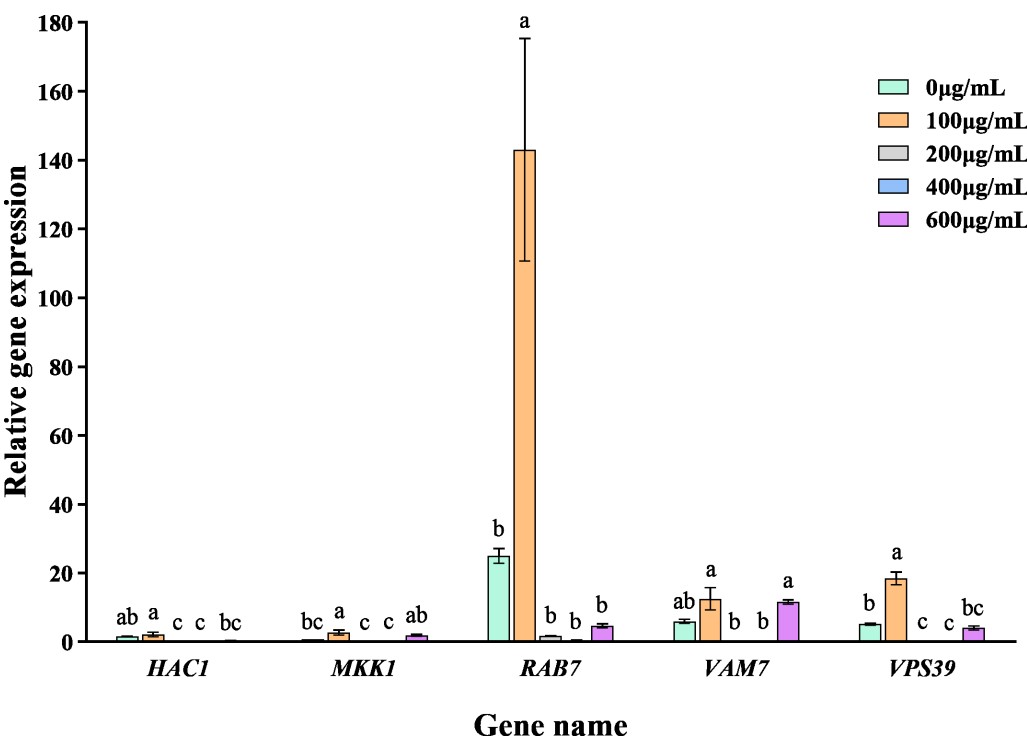

**Figure 7** **Transcriptional changes of five pathogenic genes following VFLE addition.** The values represent the means ± standard deviations. Statistical analysis was performed using $t$-test. The lowercase letters indicate a significant difference at the 0.05 level, and the same letters indicate insignificant differences.

volatiles such as DPT, DMT, and MP that proved vital in suppressing Panama banana disease in breeder practice (*Li et al., 2020*). Studies suggest that extracts from *V. fordii* leaf, flower, and husk effectively suppress anthracnose from *C. oleifera*, with leaf extracts showing enhanced efficacy (*Liu et al., 2021*). The *V. fordii* leaf-shedding phase was consistent with anthracnose infection peaks in *C. oleifera*, with interplantation significantly improving *C. oleifera* productivity compared to monoculture. In addition, intercropped forests were less prone to anthracnose infection than monocultural forests (*Zhang et al., 2019*). The abundant alkaloids in *V. fordii* leaves inhibited anthracnose, with stachydrine and betaine generally more potent against *C. fructicola*. Although the precise role of alkaloid volatiles has yet to be fully elucidated, their potential significance merits further comprehensive investigation.

Alkaloids are a diverse class, essential for biological activities, especially those related to antifungal activity. *Luo et al.*'s (*2020*) meticulous studies have definitively established the remarkable antifungal properties of Coptis alkaloids, particularly their inhibitory effects on *Trichophyton rubrum* and *Trichophyton mentagrophyte*. This antifungal activity is specifically mediated by magnoflorin, which underscores the significant potential of these alkaloids in the treatment and prevention of fungal infections (*Luo et al., 2020*). Magnoflorine served as a potent suppressor of *T. rubrum* hyphal membrane integrity, neurotransmission, and cell proliferation. In contrast, isoquinoline alkaloids showed

optimal inhibitory action against *Magnaporthe oryzae*. After treatment with sanguinarine, *M. oryzae* hyphal cell turgor was depleted, leading to bending, collapse of mycelia, and eventually damage to the cell membrane. In addition, reactive oxygen species production, mitochondrial membrane potential, and nuclear morphometry declined, ultimately impairing membrane function and hyphal proliferation (*Zhao et al., 2019*). The results of this study show that increased extracellular alkaline protease activity, malonaldehyde content, ROS content, and extracellular relative electrical conductivity of *C. fructicola* were reported at different VFLE concentrations, suggesting detrimental effects on cell wall integrity and irreversible DNA, protein, and cell membrane damage leading to pathogen cell death. Studies have highlighted the significance of CfRAB7 proteins, CfHAC1, CfVAM7, CfMKK1, and CfVPS39 in *C. fructicola* pathogenesis (*Wu, Li & Li, 2022*; *Yao et al., 2019*). Furthermore, ΔCfrab7 exerted enhanced sensitivity to $H_2O_2$ stress, demonstrating *CfRAB7* involvement in the ROS response of fruit anthrax (*Wu, Li & Li, 2022*). Increased ROS indices lead to increased levels of expression of these pathogenicity genes, particularly *CfRAB7*, in the early stages. A modest dose increased CfRAB7 protein activity while consistently high doses resulted in a significant reduction in the *CfRAB7* gene and reduced pathogenicity. The inability of cells expressing high concentrations of VFLE to penetrate cellophane supports impairment in VFLE-induced *C. fructicola* pathogen production.

Additionally, $H_2O_2$ regulates gene expression *via* redox-based epigenetic modification (*Bosch-Presegue et al., 2011*; *Cyr & Domann, 2011*; *Kreuz & Fischle, 2016*), causing transcriptional activation of redox-sensitive transcription factors like AP-1, and NRF2, CREB, HSF1, HIF-1, TP53, NF-$\kappa$B, NOTCH, SP1, SCREB-1 and FOXO family (*Schreck, Rieber & Baeuerle, 1991*; *Allen & Tresini, 2000*; *Finkel & Holbrook, 2000*; *Bonello et al., 2007*; *Matsuzawa & Ichijo, 2008*; *Akasaki et al., 2014*; *Marinho et al., 2014*; *Espinosa-Diez et al., 2015*; *Weidinger & Kozlov, 2015*). Furthermore, $H_2O_2$ affects posttranscriptional processing by regulating both cap-dependent and cap-independent translation (*Stoneley & Willis, 2004*; *Li et al., 2010*; *Zhang et al., 2012*). Our research suggests that with an elevated concentration of more than 100 µg /mL, the expression of *HACl* in *C. fructicola* of *C. oleifera* declined, implicating that VFLE may impair CfHAC1 transcription factor activity by boosting ROS levels. *Li, Zhang & Li (2021)* identified a t-SNARE protein CfVAM7 regulated by CfHAC1, affecting *C. fructicola* growth, stress response, autophagy, and pathogenicity by controlling the hyphal vacuole fusion process. Studies further demonstrated that CfVPS39, a CfVAM7-interacting protein, was also implicated in *C. fructicola* growth, autophagy, and pathogenicity regulation (*Li, Zhang & Li, 2021*). Our data revealed that while changes in the expression of *CfVAM7* and *CfVPS39* genes lacked discernable patterns, their trend remained consistent, potentially related to the interaction between *CfVAM7* and *CfVPS39*. Although *CfMKK1* expression often fluctuates within treatment groups, it suggests that *CfMKK1* is not the primary pathogenic gene repressed by VFLE.

## CONCLUSION

Ethanol extracts from *V. fordii* flowers, leaves, and husks potently inhibit *C. fructicola*'s activity, specifically VFLE. Notably, VFLE displays the most potent antifungal effect,

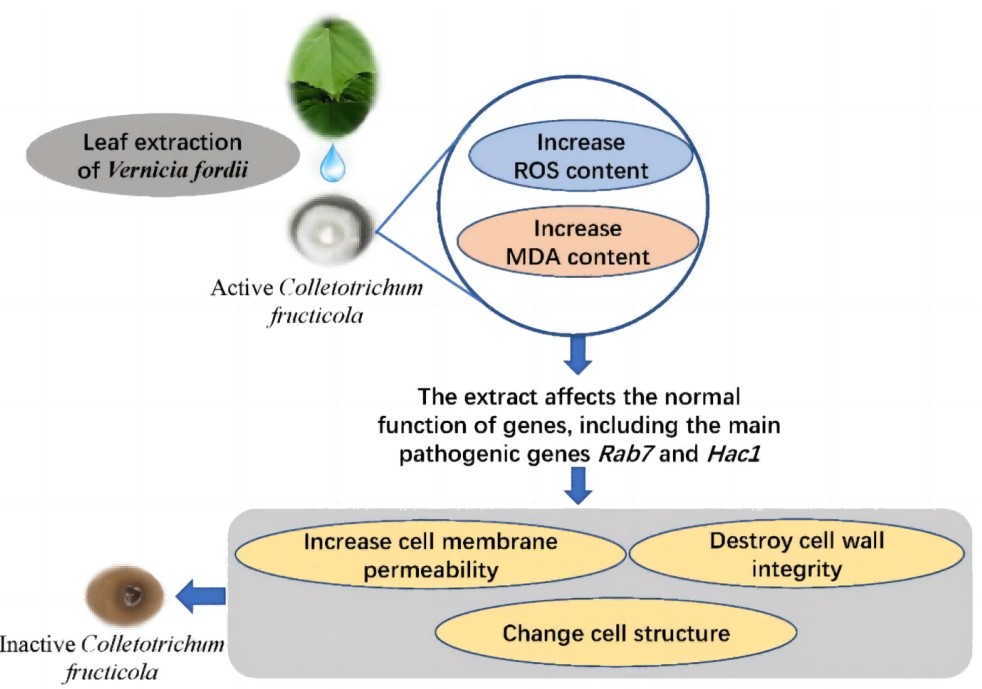

**Figure 8** **VFLE's inhibition of anthracnose in *C. oleifernia*.** The images presented in the figure, which depict *V. fordii* leaves and fungal colonies, were captured during our experiments and are original photographs. The images was taken by Yanling Zeng.

surpassing an 80% inhibition rate at a concentration of 600 μg/mL. The primary antifungal alkaloids of VFLE are hydrothylline and betaine. Experiments reveal VFLE's ability to permeate mycelium cells, elevating ROS levels and malondialdehyde content. Thus, pronounced downregulation of key pathogenic genes such as *RAB7* and *HAC1* occurs, representing the primary targets of VFLE in inhibiting anthracnose in *C. oleifera*. Further changes include increased cell membrane permeability, compromised cell wall integrity, and altered cell structure culminating in cell death (Fig. 8). These findings provide a solid scientific basis for the development of novel plant anthracnose inhibitors using *V. fordii* extracts.

### Funding

This research received funding from the Hunan Forestry Science and Technology Innovation Fund Project (XLKY202221), the National Key R&D Program of China (2017YFD0601304) and the Natural Science Foundation of Hunan Province (2022JJ30998). The funders had no role in study design, data collection and analysis, decision to publish, or preparation of the manuscript.

## Grant Disclosures

The following grant information was disclosed by the authors:
The Hunan Forestry Science and Technology Innovation Fund Project: XLKY202221.
The National Key R&D Program of China: 2017YFD0601304.
The Natural Science Foundation of Hunan Province: 2022JJ30998.

## Competing Interests

The authors declare there are no competing interests.

## Author Contributions

- Luyao Ge conceived and designed the experiments, performed the experiments, analyzed the data, prepared figures and/or tables, authored or reviewed drafts of the article, and approved the final draft.
- Yanling Zeng conceived and designed the experiments, authored or reviewed drafts of the article, and approved the final draft.
- Xinyun Liu conceived and designed the experiments, performed the experiments, prepared figures and/or tables, authored or reviewed drafts of the article, and approved the final draft.
- Xinhai Pan performed the experiments, authored or reviewed drafts of the article, search for experimental methods and provide experimental guidance, and approved the final draft.
- Guliang Yang conceived and designed the experiments, authored or reviewed drafts of the article, and approved the final draft.
- Qinhui Du performed the experiments, authored or reviewed drafts of the article, explore experimental procedures and supervise the experiment, and approved the final draft.
- Wenlin He performed the experiments, authored or reviewed drafts of the article, explore experimental procedures and supervise the experiment, and approved the final draft.

## Data Availability

The original data for all figures and tables are available in the Supplementary Files.

## Supplemental Information

Supplemental information for this article can be found online at http://dx.doi.org/10.7717/peerj.17607#supplemental-information.

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
