# Peer review of "Vernicia fordii leaf extract inhibited anthracnose growth by downregulating reactive oxygen species (ROS) levels in vitro and in vivo"

_PeerJ, doi:10.7717/peerj.17607_

## Round 0.1 · original submission · Major Revisions

The manuscript could be accepted after revision. Please pay attention to the reviewers' points of criticism.

**Language Note:** The review process has identified that the English language must be improved. PeerJ can provide language editing services - please contact us at [email protected] for pricing (be sure to provide your manuscript number and title). Alternatively, you should make your own arrangements to improve the language quality and provide details in your response letter. – PeerJ Staff

Reviewer 1 ·

Basic reporting

no comment

Experimental design

1. Line 96: I would like to ask why a 0.45 um filter was used for the plant extract instead of a 0.22 um filter, which might allow microbes to pass through. Is it possible those extracts also contain endophyte?
2. For gene expression analysis, if the experiment was only performed in liquid culture, can the gene response results infer actual pathogenicity?
3. There appears to be an inconsistency in the experimental methodology. Since the leaf extract is dissolved in DMSO, the control should also use the same concentration of DMSO to ensure comparability. However, this detail is conspicuously absent from the materials, methods, and result charts and tables, possibly affecting the reliability of the findings.

Validity of the findings

1. How did you inoculate strains on C. oleifera leaves? You should describe it in the Materials and Methods section.
2. What treatment has been applied to the C. oleifera leaves in Figure 2C? The caption description of figure 2 needs to be more accurate.

Additional comments

In conclusion, the authors' overall conclusion that Vernicia fordii leaf extract inhibits anthracnose growth in C. oleifera by downregulating ROS levels and affecting pathogenic gene expression is well-supported by the arguments, materials, analyses, and interpretations presented throughout the manuscript. The study significantly contributes to understanding the mechanisms underlying the antifungal properties of Vernicia fordii leaf extract and its potential application in managing anthracnose in C. oleifera.

Additionally, there may be some issues that could be expanded upon in future research (not necessary to reiterate in this manuscript):

Broader Sample Testing: The study might have focused primarily on a specific strain of C. fructicola. Future work could consider testing a wider array of strains to evaluate the resistance effects of Vernicia fordii leaf extract on different strains, thereby verifying its broad-spectrum antimicrobial properties.
Broader Sample Testing: The study might have focused primarily on a specific strain of C. fructicola. Future work could consider testing a wider array of strains to evaluate the resistance effects of Vernicia fordii leaf extract on different strains, thereby verifying its broad-spectrum antimicrobial properties.
Environmental Impact Assessment: The potential impact of Vernicia fordii leaf extract on the environment was not mentioned in the study. Future research could assess the eco-friendliness of this natural extract and whether its long-term use might adversely affect the structure of soil microbial communities.

Reviewer 2 ·

Basic reporting

The manuscript entitled ‘Vernicia fordii leaf extract inhibited anthracnose
growth by downregulating reactive oxygen species(ROS) levels in vitro and in vivo’ represents some useful information. The authors looked at various reactions. The title can improved accordingly.

Experimental design

It is advisable to do some tests using PDA plates (please see manuscript lines 114 and 156)

Validity of the findings

They are ok. It is advisable to do some tests using PDA plates (please see manuscript lines 114 and 156). This will enhance the manuscript.

Additional comments

It is advisable to check the English of the manuscript throughout the text. Some of the comments are placed below. It is advisable to do some tests using PDA plates (please see manuscript lines 114 and 156)
Line 16 Please change strain to species
Line 28 Please change C. fructicola mycelial,
To
C. fructicola mycelia,
Line 31 Please change
antibacterial
To antifungal
Since C. fructicola is a fungus species, not bacterium
Please improve the English of
RAB7 and HAC1 gene expression
Line 35
leaf extracts from Vernicia fordii
please change to
leaf extracts, Vernicia fordii
please improve line 43
Liu Liping's research (Liu et al., 2009) indicates that
Line 45 and Shigella sp.?
Line 49 failure rates? Please improve
Line 53 mangos? Others singular
Line 54 persimmon, so forth. Please improve
Line 61 relief. Please improve
Line 81 pulverized?
Lines 101-103 please improve English
Line 114 PDB or PDA? How it is measured? It will be useful to use PDA also.
Line 118 mycelia and conidia?
Line 118 PBS? Please write full name
Line 119 please improve the English
Line 153 please improve anti-potency
Line 155 please improve
Line 156 Only PDB used? Maybe it is better to use PDA also
Line 162 please improve
Line 170 should be V. Fordii
Line 174 please give references
Line 180 antibacterial or antifungal? bioactive compounds?
Line 189 reduction in virulence or suppressing the fungus?
Line 197 ?
Line 257 which studies, this study?
Line 261 please give reference
Line 263 alkaloid?
Line 264 please improve English
Line 266 please improve English
Lines 274-278 which study?
Line 278 this study?
Line 280 which study?
Please clearly indicate if this study or other studies
Line 297 Li Sizheng or (Li et al., 2021).?

---

## Round 0.2 · Minor Revisions

There are a few final things to ask the authors to clear up:

First, on the figures with bar graphs, the authors should explain what the a, b, c, etc relate to in terms of statistical differences. The legends say everything is <0.05, but if so, why are different letters used? Please clarify.

Figures 3 and 6 are descriptive. The authors claim to have done these experiments three times, but yet we ae given only a single descriptive figure. This may be valid, but it warrants some attention.

Table 1 could be a supplemental table, and Table 2 is brief and could be incorporated into the text.

---

## Round 0.3 · accepted · Accept

Now manuscript can be accepted